# Impact of Continuous Veno-Venous HemoDiALYsis with Regional Citrate Anticoagulation on Non-NUTRItional Calorie Balance in Patients on the ICU—The NUTRI-DAY Study

**DOI:** 10.3390/nu15010063

**Published:** 2022-12-23

**Authors:** Simon Wechselberger, Friederike Compton, Johannes Schilling

**Affiliations:** Medizinische Klinik m.S. Nephrologie und Internistische Intensivmedizin, Charité—Universitätsmedizin Berlin, 10117 Berlin, Germany

**Keywords:** acute kidney injury, continuous renal replacement therapy, nutritional support

## Abstract

Background: Malnutrition as well as overfeeding can have negative impacts on clinical outcomes in critically ill patients. Continuous veno-venous hemodialysis (CVVHD) with regional citrate anticoagulation (RCA) using trisodium citrate 4% (TSC) might play a role in nutrient disposition in patients in the ICU. Methods: In 33 consecutive patients on CVVHD with RCA, energy uptake or loss was calculated. Three macronutrients (lactate, glucose and citrate) were analyzed by taking prefilter blood and effluent samples. Results: Glucose and lactate clearance through CVVHD made up for a loss of 61 kcal/d (IQR 25–164 kcal/d) and 38 kcal/d (IQR 23–59 kcal/d), respectively. Two patients with hyperglycemic state (>350 mg/dL) lost around 600 kcal/d during CVVHD. Net post-filter citrate caloric delivery through RCA was 135 kcal/d (IQR: 124–144 kcal/d). Adding the three macronutrients, net caloric gain through CVVHD was 10 kcal/d (IQR: −63–75 kcal/d). Conclusion: In non-hyperglycemic patients on CVVHD with RCA, the metabolic contribution of the three macronutrients lactate, glucose and citrate is neglectable.

## 1. Introduction

Continuous renal replacement therapy (CRRT) is a well-established procedure for hemodynamically unstable patients with acute kidney injury (AKI) in the intensive care unit (ICU). Depending on the type of CRRT and anticoagulation modality employed, CRRT can be a potential source of macronutrient uptake or loss. Citrate for the purpose of regional citrate anticoagulation (RCA) as well as glucose and lactate as parts of RCA-solution, dialysate or replacement fluid can play a role in nutrient disposition in patients on CRRT [1,2,3,4,5]. Current guidelines on clinical nutrition in the ICU recommend that non-nutritional calorie intake through CRRT should be taken into account [6]. Previous studies have primarily investigated the metabolic contribution of continuous veno-venous hemofiltration (CVVH) and continuous veno-venous hemodiafiltration (CVVHDF), reporting substantial daily caloric gains of 500–1300 kcal [2,3,5]. To the best of our knowledge, until today, net calorie supply in patients on continuous veno-venous hemodialysis (CVVHD) with trisodium citrate 4% (TSC)-RCA remains unclear. Our study aimed to estimate the impact of CVVHD with TSC-RCA on energy balance in patients in the ICU. We further suggest an easy to obtain bedside equation for everyday calculation of calorie supply from CVVHD.

## 2. Materials and Methods

### 2.1. Patient Population

An investigator initiated, non-randomized, single-center, retrospective study including consecutive adult patients undergoing CVVHD with TSC-RCA on an internal medicine ICU was conducted between March 2021 and September 2021. There were no further exclusion criteria.

The study was approved by the institutional ethical review board (“Impact of continuous veno-venous hemoDiAlYsis with regional citrate anticoagulation on non-NUTRItional calorie balance in patients on the ICU—the NUTRI-DAY study”, EA4/200/21, Ethikkommission der Charité, 3 November 2021), and conforms to the principles outlined in the Declaration of Helsinki. All data were collected, managed and analyzed at Charité university hospital.

### 2.2. CVVHD Protocol

CVVHD was initiated and further managed per institutional protocol and was performed using the Fresenius Multifiltrate platform (Fresenius Medical Care, Bad Homburg, Germany) and AV1000S-dialyzer (Polysulfone, surface area 1.8 m^2^, Fresenius Medical Care, Bad Homburg, Germany). A fixed blood flow rate/dialysis solution flow rate ratio of 100 (mL/min)/2000 (mL/h) or 150 (mL/min)/3000 (mL/h) was used, respectively, exceeding an administered effluent volume of 20 mL/kg/h, as recommended by the Kidney Disease: Improving Global Outcomes (KDIGO) guidelines [7]. TSC 4% was started with 4 mmol/L blood flow, corresponding with a citrate infusion rate of 176 mL/h for a blood flow rate of 100 mL/min and 264 mL/h for a blood flow rate of 150 mL/min, respectively. TSC was then titrated to reach postfilter ionized calcium blood levels between 0.25 mmol/L and 0.34 mmol/L. The postfilter supplementation of 10% calcium chloride (CaCl_2_) solution was started with 1.7 mmol/L dialysate flow, corresponding with a calcium infusion rate of 39 mL/h for a dialysis solution flow rate of 2000 mL/h and 58.5 mL/h for a dialysis solution flow rate of 3000 mL/h, respectively, and was further titrated to reach systemic ionized calcium levels between 1.12 and 1.20 mmol/L. Calcium-free dialysate (Ci-Ca^®^ Dialysate K2, Fresenius Medical Care, Bad Homburg, Germany, Table 1) was used and the ultrafiltration rate was modified according to the hemodynamic needs of the patients. The standard temperature of the dialysate was 37 °C. An illustration of the CVVHD set-up is shown in Figure 1.

### 2.3. Data Protocol

Next to baseline demographic data (age, gender, height, weight), nutrition administration (enteral, parenteral, none) and SOFA Score, CVVHD parameters [TSC dosage (mmol/L), CaCl_2_ dosage (mmol/L), dialysis solution flow rate (mL/h), blood flow rate (mL/min), and ultrafiltration/24 h] were collected.

Between 12 and 72 h after the start of the CVVHD, prefilter blood and effluent samples were taken simultaneously to measure the concentrations of blood glucose, blood lactate, hemoglobin, hematocrit (hct), effluent glucose, effluent lactate and effluent citrate.

Prefilter blood glucose and lactate concentrations, as well as effluent glucose and lactate concentrations, were measured using an enzymatic amperiometric method (ABL 800flex, Radiometer, Copenhagen, Denmark). Effluent citrate concentration was measured using photometric analysis via a commercial laboratory (Labor Augsburg MVZ GmbH, Augsburg, Germany). Laboratory analyses of lactate, glucose and citrate were performed at room temperature.

### 2.4. Calculations of Lactate, Glucose and Citrate Balance

To estimate the energy balance of lactate, glucose and citrate in CVVHD, three basic formulas were used.

The caloric equivalents of the three macronutrients are 3.62 kcal per gram lactate, 3.75 kcal per gram glucose and 2.47 kcal per gram citrate [8].

The net daily lactate balance [LB, (g/24 h)] was estimated by the multiplication of daily effluent volume with lactate concentration in effluent [LE, (g/L)]. The daily effluent volume was calculated by adding dialysis solution flow rate [DF, (L/24 h)] and ultrafiltration [UF, (L/24 h)].
$$\mathrm{LB}=\left(\mathrm{DF}\;+\;\mathrm{UF}\right)\;*\;\mathrm{LE}$$

The net daily glucose balance [GB, (g/24 h)] was determined by the daily amount of glucose in effluent subtracted by the daily application of glucose in dialysate. Similar to lactate, for quantification of daily glucose in effluent, daily effluent volume was multiplicated with glucose concentration in effluent [GE, (g/L)]. The daily application of glucose in dialysate was determined by the multiplication of DF and glucose concentration in dialysate [GD, (g/L)].
$$\mathrm{GB}=\left[\left(\mathrm{DF}\;+\;\mathrm{UF}\right)\;*\;\mathrm{GE}\right]-\left[\mathrm{DF}\;*\;\mathrm{GD}\right]$$

Net daily citrate balance [CB, (g/24 h)], i.e., net post-filter citrate delivery, was estimated by subtracting citrate loss via the dialysis filter membrane from prefilter citrate supply through TSC. Citrate loss was calculated, analogous to lactate and glucose, by multiplying daily effluent volume with citrate concentration in effluent [CE (g/L)]. The prefilter citrate supply was calculated by multiplying the citrate dosage [CD (g/L)] and blood flow rate [BF (L/24 h)]. The molar mass of citrate is 192.13 g [9].
$$\mathrm{CB}=\left[\mathrm{CD}*\mathrm{BF}\right]-\left[\left(\mathrm{DF}+\mathrm{UF}\right)*\mathrm{CE}\right]$$

Citrate concentration per plasma volume (mmol/L) was calculated by dividing citrate dosage in mmol/L by 1-(hct/100).

The proportion (%) of citrate clearance was calculated by dividing citrate in effluent (kcal) by prefilter citrate supply (kcal).

### 2.5. Statistics

Categorical variables were reported as count and percentages and continuous variables as mean and standard deviation if normally distributed, otherwise as median with interquartile range.

A Mann-Whitney U test was used to compare the percentages of citrate clearance between patients with and without ultrafiltration.

Univariate linear regression models were used to predict the values of effluent lactate based on blood lactate, effluent glucose based on blood glucose, and effluent citrate based on plasma citrate, respectively.

All statistical analyses were performed using SPSS 28. A *p*-value of <0.05 was considered statistically significant.

## 3. Results

### 3.1. Study Cohort

A total of 33 consecutive patients were included in the study. Baseline characteristics are shown in Table 2. Most of the patients (90%) were treated with a blood flow rate of 100 mL/min, and a dialysis solution flow rate of 2000 mL/h. Ten percent had a blood flow rate of 150 mL/min and a dialysis solution flow rate of 3000 mL/h. The median prescribed dialysate solution flow rate was 24.7 mL/kg/h (IQR 22.2–28.6 mL/kg/h). In 61% of the patients, no ultrafiltration was applied, whereas 39% of the patients were treated with a median ultrafiltration volume of 2.2 L/d (IQR 1.1–2.4). The TSC infusion rate was 4 mmol/L ± 0.09 mmol/L. The postfilter CaCl_2_ infusion rate was 1.8 mmol/L ± 0.2 mmol/L.

### 3.2. Lactate Caloric Balance

The median loss of calories due to lactate clearance during CVVHD was 38 kcal/d (IQR 23–59 kcal/d) (Figure 2). One patient had a highly elevated blood lactate concentration (141 mg/dL) resulting in a loss of 258 kcal/d due to lactate clearance on CVVHD.

Median lactate levels in prefilter blood (19 mg/dL, IQR: 13–33 mg/dL) and effluent (19 mg/dL, IQR 13–31 mg/dL) showed no relevant difference from each other and blood lactate concentration was a very strong predictor for effluent lactate concentration in the linear regression model (R^2^ = 0.98, *p* < 0.001, Figure 3).

### 3.3. Glucose Caloric Balance

The median loss of glucose during CVVHD was 61 kcal/d (IQR 25–164 kcal/d) (Figure 2). Two patients lost more than 600 kcal/d of glucose, and both of them had blood glucose levels greater than 350 mg/dL. One of these patients was hyperglycemic because of a hyperosmolar hyperglycemic state, and the other was treated with high doses of glucose due to high-dose insulin euglycemic therapy applied for severe calcium channel blocker poisoning.

The median concentration of prefilter blood glucose was 131 mg/dL (IQR 115–186 mg/dL) and the median effluent glucose concentration was 130 mg/dL (IQR 113–185 mg/dL), revealing no relevant difference in these two concentrations. Blood glucose concentration was a strong predictor for effluent glucose concentration displaying greater loss of glucose at higher blood glucose dialysate glucose gradients (R^2^ = 0.99, *p* < 0.001, Figure 3).

### 3.4. Citrate Caloric Balance

RCA with TSC accounted for a median prefilter citrate supply of 271 kcal/d (IQR 271–277 kcal/d). About half of the citrate (Median: 53%, IQR 48–57%) was cleared during the process of CVVHD, which resulted in a net median citrate caloric delivery of 135 kcal/d (IQR 124–144 kcal/d) (Figure 2).

Since citrate distributes only in plasma, citrate concentration per plasma volume was calculated. Median hematocrit was 27% (IQR 25–34%). Median citrate concentration in plasma was 5.6 mmol/L (IQR 5.3–6.2), whereas median citrate concentration in effluent was 6.28 mmol/L (IQR 5.72–6.72 mmol/L), respectively. Calculated citrate concentration per plasma volume appeared to be a predictor for citrate concentration in effluent in a linear regression model (R^2^ = 0.32, *p* < 0.001, Figure 3). There was no significant difference for the proportion of citrate clearance between patients with ultrafiltration (Median: 54%, IQR: 50–56%) and patients without ultrafiltration (Median: 52%, IQR: 47–57%), respectively (*p* = 0.43).

### 3.5. Net Caloric Balance

Adding the three observed macronutrients, the median net caloric gain through CVVHD was 10 kcal/d (IQR −63–75 kcal/d) (Figure 3). Two patients lost around 600 kcal/d during the treatment. Both of these patients had high glucose blood levels in particular, resulting in a major loss of glucose calories due to CVVHD.

## 4. Discussion

Malnutrition and overfeeding can have a negative impact on clinical outcomes in critically ill patients [10,11,12]. Even though current guidelines on nutrition in the ICU recommend that the calorie supply from CVVHD should be taken into account, evidence on the bioenergetic gains or losses due to the procedure is lacking. While previous studies have primarily investigated the caloric impact of CVVH and CVVHDF, we conducted the first tests on patients on CVVHD with TSC-RCA. Our findings do not confirm the older results of major caloric uptake through CRRT, but instead illustrate caloric neutrality and in certain situations (i.e., hyperglycemia, lactate-acidosis) even negative bioenergetic impact of CVVHD. Non-nutritional energy supply from CVVHD with TSC-RCA is mostly driven by citrate delivery through TSC, whereas energy loss is proportionally related to elevated blood glucose and, to a lower extent, lactate concentrations.

We propose the first predictive bedside equation for the everyday calculation of caloric contribution of CVVHD. The equation can be applied only by using routine point of care tests without the need of any additional laboratory measurements.

### 4.1. Lactate Balance

In modern CVVHD, standard dialysate contains no or only little lactate [13]. Due to low molecular weight, lactate freely transfers across the dialysis filter membrane. As a result of lactate free dialysate in our study, a net loss of lactate was observed in all of our patients. Nevertheless, for patients with non-elevated blood lactate concentrations, the impact of caloric loss due to lactate clearance is rather small. For example, patients in our cohort with non-elevated blood lactate concentrations (≤20 mg/dL) did not lose more than 55 kcal due to lactate clearance on CVVHD. Older studies examining lactate buffered solutions as dialysate and replacement fluids in CVVH and CVVHDF reported calorie gains of 400–500 kcal/d [2,3].

### 4.2. Glucose Balance

In CVVHD with TSC-RCA, glucose uptake or loss is, similar to lactate, fundamentally dependent on blood glucose concentrations. Glucose concentration in dialysate was 100 mg/dL. Blood glucose concentration targets for critically ill patients on ICU are recommended to be between 150 mg/dL and 180 mg/dL [14]. This leads to a glucose gradient towards the dialysate and a net loss of glucose calories on CVVHD in most patients. In our cohort, the median prefilter blood glucose concentration was 131 mg/dL, resulting in a net loss of around 61 kcal/d. Thus, patients with rather high prefilter glucose levels lost relevant amounts of calories from CVVHD. For example, one patient with a blood glucose level of 431 mg/dL lost a total of 607 kcal/d. Glucose free dialysate would even exaggerate this effect [1]. On the other hand, patients with blood glucose concentrations less than 100 mg/dL could even gain calories due to CVVHD. Depending on the modality of CRRT, the dialysate and/or replacement fluids and the components of RCA, energy supply from glucose can widely differ. Previous studies have primarily investigated the metabolic contribution of glucose in CVVH and CVVHDF, reporting a net caloric supply of 300–600 kcal/d [2,3,5]. Those studies have predominantly used acid citrate dextrose, formula-A (ACD-A) for RCA, which contains 138.9 mmol glucose per liter, instead of glucose free TSC, explaining the positive glucose energy balance.

### 4.3. Citrate Balance

Net non-nutritional calorie supply through citrate in CVVHD is dependent on the amount of prefilter TSC infusion as well as elimination via the dialysis filter membrane.

The prefilter TSC concentration is adjusted upon post-filter ionized blood calcium measurements and the TSC infusion rate is directly linked to the blood flow rate. The median prefilter caloric supply via prefilter TSC infusion made up for a total of 271 kcal/d in our cohort.

Citrate- and calcium free dialysate leads to citrate diffusion across the dialysis filter membrane. Since the absolute blood flow rate in CVVHD is always higher than the absolute dialysis solution flow rate, complete citrate saturation of the effluent is obligatory. We found calculated citrate per plasma volume to be a predictor of citrate elimination, since citrate only distributes in plasma. Patients with higher hematocrit showed higher calculated citrate concentrations per plasma, resulting in a greater absolute elimination of citrate.

With a fixed blood flow rate/dialysis solution flow rate ratio, 53% of prefilter citrate supply through TSC infusion was eliminated into the effluent, making up for a net citrate delivery of 135 kcal/d. Interestingly, patients with programmed ultrafiltration did not show significantly higher percentages of citrate clearance, which is mostly explained by the rather low proportions of ultrafiltration regarding the total effluent volume.

Previous studies have mostly assessed the effect of citrate delivery through ACD-A in patients on CVVH and CVVHDF. ACD-A contains 113 mmol citrate/L, whereas TSC contains 136 mmol citrate/L. Nevertheless, those studies reported net citrate calorie gains of around 200–500 kcal/d [2,3,5,15]. Absolute citrate dosing and delivery to patient’s metabolism was higher than with TSC in our study, mostly due to higher blood flow rates, calcium containing replacement fluids and calcium containing dialysate, leading to higher citrate infusion rates to maintain postfilter calcium targets. As a result, higher bioenergetic impact of RCA has been observed.

### 4.4. Net Balance for Lactate, Glucose, Citrate and Bedside Equation

While previous studies have discussed the risk of unrecognized exogenous energy delivery from CRRT (500–1300 kcal/d), we could show that in modern CVVHD with TSC-RCA, the caloric uptake and loss is mostly balanced. Minor losses of calories due to the clearance of lactate and glucose are in most cases compensated for by citrate delivery from TSC-RCA. Nevertheless, patients with substantially elevated glucose and/or lactate concentrations do experience losses of calories due to high gradients towards the dialysis solution. Even though the clinical relevance of this exogenous malnutrition has to be individually assessed, clinicians should get a sense of this phenomenon.

The utilization of a user friendly predictive bedside equation for clinicians to estimate the impact of CVVHD with TSC-RCA on energy balance seems reasonable. Our findings lead to fundamental simplifications in the calculation of bioenergetic impact of three macronutrients in CVVHD. First, blood glucose and blood lactate concentrations showed no relevant differences from effluent glucose and effluent lactate concentrations, respectively. Therefore, they might be used as surrogate parameters for effluent glucose and lactate concentrations in the predictive bedside equation, and testing of effluent levels can be waived. Due to possible changes in blood glucose and blood lactate concentrations, we suggest to use average daily blood glucose and blood lactate concentrations for calculations. Second, our data illustrated that around half of the applied prefilter citrate (53%) is eliminated over the dialysis filter membrane into the effluent. For everyday calculations in a clinical setting, we propose multiplying the prefilter citrate dosage by 0.5 (approximative factor of citrate elimination). Measurements of citrate in effluent become redundant. To sum up, the energy balance of three macronutrients in CVVHD can be calculated only by applying routinely taken point of care tests and employed CVVHD modalities.

Examples of calculated net calorie balances per day (kcal/d) using the predictive bedside equation for different glucose and lactate concentrations, different blood flow and dialysis solution flow rates, respectively, with exemplary citrate dosage of 4 mmol/L, and no ultrafiltration, are illustrated in Table 3 and Table 4.
$$\begin{array}{c} Net\;caloric\;balance=citrate\;dosage\;\left(\frac{g}{L}\right)*\;blood\;flow\;\left(\frac{L}{d}\right)*\;2.47\;\left(\frac{kcal}{g}\right)*\;0.5\\ -\;[\left(dialysate\;flow\;rate\;\left(L/d\right)+ultrafiltration\;\left(L/d\right)\right]*avg.\;lactate\;blood\;\left(g/L\right)*\;3.62\;\left(kcal/g\right)\\ -[[\left(dialysate\;flow\;rate\;\left(\frac{L}{d}\right)+ultrafiltration\;\left(\frac{L}{d}\right)\right)*\;avg.\;glucose\;blood\;\left(\frac{g}{L}\right)]-\;[dialysate\;flow\;rate\;\left(\frac{L}{d}\right)\\ {}*\;glucose\;dialysate\;\left(\frac{g}{L}\right)]]*\;3.75\left(\frac{kcal}{g}\right) \end{array}$$

### 4.5. Limitations

This is a non-randomized observational study. Our data were collected in a cohort of patients being treated in an internal medicine ICU with a standard CVVHD and TSC-RCA protocol and cannot be generalized to other groups of patients and treatment protocols.

Since blood and effluent samples were taken only once, the calculated daily calorie load can only be seen as an estimate.

By using a standard, operator instruction guided protocol of CVVHD with TSC-RCA with a fixed blood flow/dialysis solution flow ratio, no effect on citrate clearance regarding the change of blood flow/dialysis solution flow ratio could be revealed. In patients with metabolic acidosis, a higher ratio is sometimes applied to reduce absolute citrate clearance, since citrate is metabolized to bicarbonate [16]. This would, as a side effect, result in a higher calorie gain through TSC-RCA.

In 85% of the patients, the effluent citrate concentration was higher than the calculated plasma citrate concentration. This phenomenon might predominantly be due to the re-circulation effects of citrate during CVVHD. Patients’ organs do not metabolize all of the citrate before blood reaches the dialysis-cycle again [5,17]. In addition, there are small endogenous physiologic citrate concentrations not being generated by the TSC-RCA [2]. These aspects lead to the minor underestimation of calculated prefilter plasma citrate concentrations.

The bioenergetic and hydric body balances by real and undetectable perspiration and breathing were not considered.

## 5. Conclusion

An analysis of the three macronutrients of lactate, glucose and citrate on CVVHD with TSC-RCA revealed no relevant effect on bioenergetic uptake or loss in patients with non-elevated blood glucose and lactate concentrations. However, patients with elevated glucose and/or lactate concentration are predestined to lose relevant numbers of calories on CVVHD.

## Figures and Tables

**Figure 1 nutrients-15-00063-f001:**
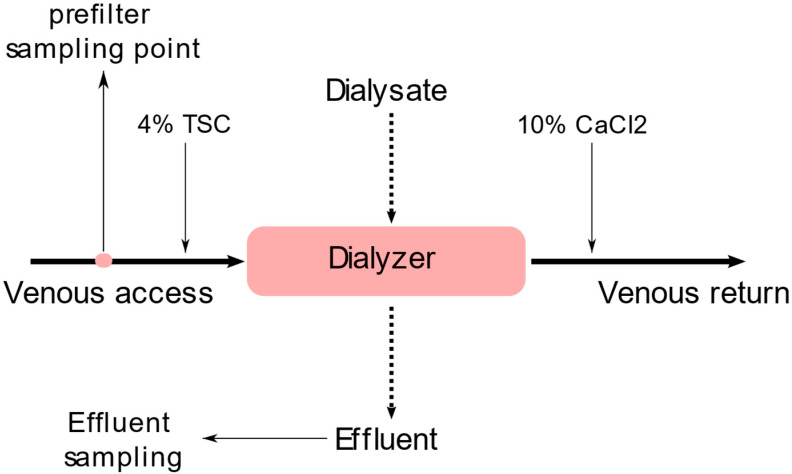
CVVHD set up and sampling points.

**Figure 2 nutrients-15-00063-f002:**
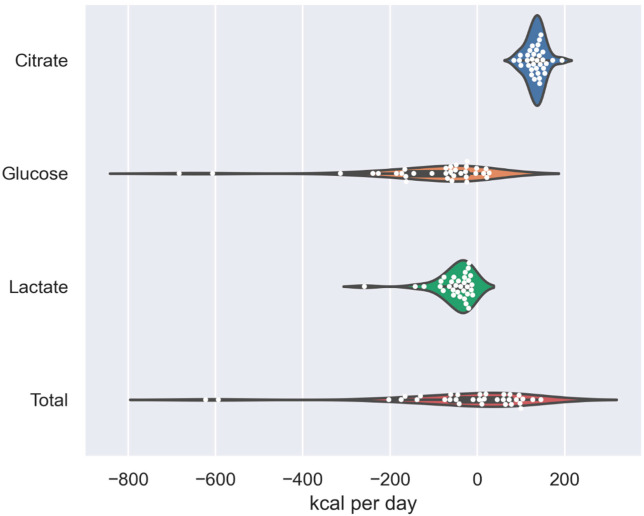
Violin plot of caloric balances of the three macronutrients (citrate, glucose, lactate) and total calorie balance.

**Figure 3 nutrients-15-00063-f003:**
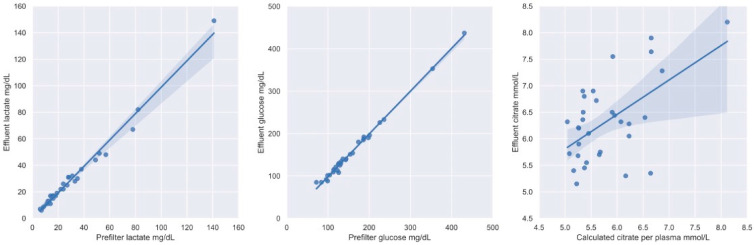
Linear regression models with 95% confidence intervals of prefilter lactate and lactate in effluent, prefilter glucose and glucose in effluent, and calculated citrate per plasma and citrate in effluent, respectively.

**Table 1 nutrients-15-00063-t001:** Content of dialysate.

Content	Ci-Ca^®^ Dialysate K2
Sodium, mmol/L	133.0
Potassium, mmol/L	2.0
Chloride, mmol/L	116.5
Sodium bicarbonate, mmol/L	20.0
Magnesium, mmol/L	0.75
Glucose, g/L	1.0

**Table 2 nutrients-15-00063-t002:** Patient characteristics.

Characteristic	Value
Age (years)	67 ± 14
Men/Women (n)	24/9
Body weight (kg)	86 ± 24
BMI (kg/m^2^)	29 ± 8
Nutrition administration (n)	
• oral/enteral • parenteral • none	20 3 10
SOFA Score	12 ± 4

**Table 3 nutrients-15-00063-t003:** Calculated net calorie balance per day (kcal/d) from CVVHD for different blood glucose (100–500 mg/dL) and lactate (10–100 mg/dL) concentrations with exemplary citrate dosage of 4 mmol/L, blood flow rate of 100 mL/min., dialysis solution flow rate of 2000 mL/h, and no ultrafiltration.

		Glucose mg/dL								
		100	150	200	250	300	350	400	450	500
**Lactate mg/dL**	**10**	119.6	29.6	−60.4	−150.4	−240.4	−330.4	−420.4	−510.4	−600.4
	**20**	102.2	12.2	−77.8	−167.8	−257.8	−347.8	−437.8	−527.8	−617.8
	**30**	84.9	−5.1	−95.1	−185.1	−275.1	−365.1	−455.1	−545.1	−635.1
	**40**	67.5	−22.5	−112.5	−202.5	−292.5	−382.5	−472.5	−562.5	−652.5
	**50**	50.1	−39.9	−129.9	−219.9	−309.9	−399.9	−489.9	−579.9	−669.9
	**60**	32.7	−57.3	−147.3	−237.3	−327.3	−417.3	−507.3	−597.3	−687.3
	**70**	15.4	−74.6	−164.6	−254.6	−344.6	−434.6	−524.6	−614.6	−704.6
	**80**	−2.0	−92.0	−182.0	−272.0	−362.0	−452.0	−542.0	−632.0	−722.0
	**90**	−19.4	−109.4	−199.4	−289.4	−379.4	−469.4	−559.4	−649.4	−739.4
	**100**	−36.8	−126.8	−216.8	−306.8	−396.8	−486.8	−576.8	−666.8	−756.8

**Table 4 nutrients-15-00063-t004:** Calculated net calorie balance per day (kcal/d) from CVVHD for different blood glucose (100–500 mg/dL) and lactate (10–100 mg/dL) concentrations with an exemplary citrate dosage of 4 mmol/L, blood flow rate of 150 mL/min., dialysis solution flow rate of 3000 mL/h, and no ultrafiltration.

		Glucose mg/dL								
		100	150	200	250	300	350	400	450	500
**Lactate mg/dL**	**10**	178.9	43.9	−91.1	−226.1	−361.1	−496.1	−631.1	−766.1	−901.1
	**20**	152.9	17.9	−117.1	−252.1	−387.1	−522.1	−657.1	−792.1	−927.1
	**30**	126.8	−8.2	−143.2	−278.2	−413.2	−548.2	−683.2	−818.2	−953.2
	**40**	100.7	−34.3	−169.3	−304.3	−439.3	−574.3	−709.3	−844.3	−979.3
	**50**	74.7	−60.3	−195.3	−330.3	−465.3	−600.3	−735.3	−870.3	−1005.3
	**60**	48.6	−86.4	−221.4	−356.4	−491.4	−626.4	−761.4	−896.4	−1031.4
	**70**	22.6	−112.4	−247.4	−382.4	−517.4	−652.4	−787.4	−922.4	−1057.4
	**80**	−3.5	−138.5	−273.5	−408.5	−543.5	−678.5	−813.5	−948.5	−1083.5
	**90**	−29.6	−164.6	−299.6	−434.6	−569.6	−704.6	−839.6	−974.6	−1109.6
	**100**	−55.6	−190.6	−325.6	−460.6	−595.6	−730.6	−865.6	−1000.6	−1135.6

## Data Availability

The dataset used and analyzed during the current study is available from the corresponding author on reasonable request.

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
