# Peer review of "Impact of Continuous Veno-Venous HemoDiALYsis with Regional Citrate Anticoagulation on Non-NUTRItional Calorie Balance in Patients on the ICU—The NUTRI-DAY Study"

_nutrients, 2022, doi:10.3390/nu15010063_

Round 1

Reviewer 1 Report

In this manuscript, the authors explored the impact of citrate anticoagulation on calorie balance in patients under CVVHD. In summary, CVVHD had subtle impact on carbohydrate calory balance due to the exchange of lactate, glucose and citrate except for patients with high serum lactate or glucose. This manuscript is well-written and provide an important information in nutritional care in intensive care unit. I have several suggestions as follows.

Major point:

・The primary result of this study is caloric balances of lactate, glucose and citrate. The results were described in the beginning of each paragraph in Results. However, its figure (Figure 3) appeared after the linear regression model (Figure 2) which is complementary to the caloric balance. I suggest revise the order of the figures.

 Additionally, the description of statistics according to the regression models were too simple (Lines 120–121). It is better to describe in detail what relationship was investigated using linear regression model.

・Informed consent can be waived in this retrospective study. However, opt-out should be disclosed.

Minor points:

・In Discussion, the authors are discussing caloric balance from “carbohydrate”. Macro nutrients such as amino acids are also excreted to dialysate. I recommend authors to avoid the description which may lead readers to misunderstand that the authors are talking about total energy balance (ex. Paragraph “Net caloric balance and bedside equation”)

・The explanation abbreviation “TSC” is not written in the main text.

Author Response

Response to Reviewer 1 Comments

Point 1: The primary result of this study is caloric balances of lactate, glucose and citrate. The results were described in the beginning of each paragraph in Results. However, its figure (Figure 3) appeared after the linear regression model (Figure 2) which is complementary to the caloric balance. I suggest revise the order of the figures.

Response 1: Thank you very much for reviewing our manuscript. We appreciate your feedback. We have revised the order of the figures according to your comment.

Point 2: Additionally, the description of statistics according to the regression models were too simple (Lines 120–121). It is better to describe in detail what relationship was investigated using linear regression model.

Response 2: We have updated the methods according to this proposal.

Point 3: Informed consent can be waived in this retrospective study. However, opt-out should be disclosed.

Response 3: Opt-out recruitment approach was assessed. We have added this statement to the manuscript.

Point 4: In Discussion, the authors are discussing caloric balance from “carbohydrate”. Macro nutrients such as amino acids are also excreted to dialysate. I recommend authors to avoid the description which may lead readers to misunderstand that the authors are talking about total energy balance (ex. Paragraph “Net caloric balance and bedside equation”)

Response 4: We have adapted the manuscript to avoid missunderstandings about amino acid calorie balance in CVVHD, which was not part of our study.

Point 5: The explanation abbreviation “TSC” is not written in the main text.

Response 5: We have added the explanation in the main text.

Reviewer 2 Report

Overall, this is a well written article. However, the authors would have to describe the unique contribution of this study to the current literature. A more detailed description of the methodology and statistical analysis, as well as the discussion, is required. 

Author Response

Response to Reviewer 2 Comments

Point 1: Overall, this is a well written article. However, the authors would have to describe the unique contribution of this study to the current literature. A more detailed description of the methodology and statistical analysis, as well as the discussion, is required. 

Response 1: Thank you very much for reviewing our manuscript. We appreciate your feedback. We have adopted all sections of the manuscript according to your review.

Our study gives important evidence on calorie balance in patients on CVVHD with TSC-RCA. ESPEN guideline on clinical nutrition in the intensive care unit recommend that non nutritional calorie intake through CRRT should be taken into account. While previous studies have primarily investigated the calorie impact of CVVH and CVVHDF, we conduct the first results on CVVHD in the biggest cohort of any of those studies so far. Our findings do not confirm older results of major caloric uptake through CRRT, instead show caloric neutrality and in certain situations even caloric loss on CVVHD with TSC. Our findings might have a future impact when it comes to choosing the most fitting type of CRRT in the context of energy supply. Finally we conduct the first bedside equation for everyday calculation of caloric contribution of CVVHD. The equation is very user friendly because it can be applied without the need of any extra testing apart from routinely taken point of care tests.

Reviewer 3 Report

This is an interesting analysis of calorie balance with CRRT using citrate for regional anticoagulation.

This paper suggests that CRRT with citrate is practically calorie-neutral, except in patients with significant hyperglycemia or lactic acidosis.

One specific comment regarding Figure 2 and the text in the body of the paper associated with it:  The graphs show pre-filter versus dialysate.   I believe that "Dialysate" should be changed to "Effluent," if I'm understanding it correctly.

The other suggestion I would make is that the equation you present in the discussion is not simple.  It  might be more clinically useful to state the calories lost per day for elevation in average blood glucose above a certain threshold, and a separate equation stating the calories lost per day for lactic acid above a specific level.

Author Response

Response to Reviewer 3 Comments

Point 1: One specific comment regarding Figure 2 and the text in the body of the paper associated with it:  The graphs show pre-filter versus dialysate.   I believe that "Dialysate" should be changed to "Effluent," if I'm understanding it correctly.

Response 1: Thank you very much for reviewing our manuscript. We appreciate your feedback. Especially thank you for your comment on the figure. We have changed the figure legend from dialysate to effluent.

Point 2: The other suggestion I would make is that the equation you present in the discussion is not simple. It might be more clinically useful to state the calories lost per day for elevation in average blood glucose above a certain threshold, and a separate equation stating the calories lost per day for lactic acid above a specific level.

Response 2: This is a very interesting suggestion. We have included two new tables with examples of calculated net calorie balances per day for different glucose and lactate concentrations, different blood flow and dialysis solution flow rates, respectively, with exemplary citrate dosage of 4 mmol/L, and no applied ultrafiltration, using the predictive bedside equation. These two tables emphasise the clinical usefulness of the manuscript. 

We would still like to keep the equation in the manuscript, because it can guide clinicans to estimate calorie balance from CVVHD by only applying routinely taken point of care testing and CVVHD modalities by themselfes. From our point of view there are two observations that make the equation user friendly: First, blood glucose and blood lactate concentrations showed no relevant differences from effluent glucose and effluent lactate concentrations, so they can be used as surrogate parameters in the equation. Therefore, there is no need to measure effluent levels of these two macronutrients in an everyday clinical setting. Second, we could show that around 50% of prefilter citrate is eliminated over dialysis filter membrane into the effluent. This gives clinicians the chance to calculate calorie balance from citrate without measuring citrate in effluent. All in all, the net calorie balance from CVVHD can be calculated by filling in results of routine point of care tests and employed CVVHD settings. We have adapted the discussion to specify on this issue.

Round 2

Reviewer 2 Report

I think the article has gained in quality

Author Response

Thank you very much for this review.